# The cGAS Paradox: Contrasting Roles for cGAS-STING Pathway in Chromosomal Instability

**DOI:** 10.3390/cells8101228

**Published:** 2019-10-10

**Authors:** Christy Hong, Andrea E. Tijhuis, Floris Foijer

**Affiliations:** European Research Institute for the Biology of Ageing, University of Groningen, University Medical Centre Groningen, A. Deusinglaan 1, Groningen, 9713 AV, The Netherlands; c.hong@umcg.nl (C.H.); a.e.tijhuis@umcg.nl (A.E.T.)

**Keywords:** mitosis, cGAS, STING, chromosomal instability, aneuploidy

## Abstract

Chromosomal instability (CIN) is an intricate phenomenon that is often found in human cancer, characterized by persisting errors in chromosome segregation. This ongoing chromosome mis-segregation results in structural and numerical chromosomal abnormalities that have been widely described to promote tumor evolution. In addition to being a driver of tumor evolution, recent evidence demonstrates CIN to be the central node of the crosstalk between a tumor and its surrounding microenvironment, as mediated by the cGAS-STING pathway. The role that cGAS-STING signaling exerts on CIN tumors is both complex and paradoxical. On one hand, the cGAS-STING axis promotes the clearance of CIN tumors through recruitment of immune cells, thus suppressing tumor progression. On the other hand, the cGAS-STING pathway has been described to be the major regulator in the promotion of metastasis of CIN tumors. Here, we review this dual role of the cGAS-STING pathway in the context of chromosomal instability and discuss the potential therapeutic implications of cGAS-STING signaling for targeting CIN tumors.

## 1. Chromosomal Instability (CIN) in Cancer

Chromosomal instability (CIN) is a process in which chromosome missegregation persists over consecutive cell divisions [1]. CIN leads to genetic alterations, including chromosome copy number changes that help cells to adapt when faced with stresses [2,3,4]. CIN should not to be mistaken with aneuploidy. CIN is a condition as a result of which chromosome missegregation happens at increased frequency, while aneuploidy is a state describing the actual abnormalities in chromosome copy numbers. While CIN and aneuploidy often co-occur in tumors, [5] aneuploidy can also exist in the absence of CIN, a condition known as stable aneuploidy [6]. An example of stable aneuploidy occurs, for instance, in Down syndrome individuals whose cells all harbor one extra copy of chromosome 21. Similarly, some cancers display high-grade aneuploidy without a CIN phenotype and are thus defined to be stably aneuploid [6].

Many human tumors display a CIN phenotype, and the severity of the CIN phenotype correlates with the stage of the affected tumors [7]. Yet, the relation between CIN and cancer is not as straightforward as it seems. For instance, CIN is believed to drive cancer drug resistance [8,9] through the generation of new karyotypes that include whole chromosome gains or losses (numerical CIN) and structural changes to chromosomes (structural CIN) that help cancer cells to overcome the drug-imposed cytotoxic effects [9]. However, elevating the CIN rates in tumors with pre-existing CIN results in enhanced sensitivity to cytotoxic chemotherapies, such as cisplatin [10] and 5-fluorouracil (5-FU) [11], which indicates that an excessive level of CIN is toxic for cancer cells.

Whether CIN acts as an ally or a foe does not only depend on the CIN rates in the tumor, but also on the interaction between the cancer cells and their microenvironment, including the immune component. For instance, transcriptome analysis revealed that tumors with high aneuploidy show a reduced infiltration of cytotoxic immune cells, in particular CD8^+^ T-cells [12] as compared to non-aneuploid tumors. Another study analyzing more than 10,000 human cancers that were deposited at The Cancer Genome Atlas (TCGA) showed a negative correlation between aneuploidy and leukocyte markers, providing further evidence that CIN tumors need to duck the immune system [13]. The molecular mechanism connecting a CIN phenotype to the immune system appears to be cyclic GMP AMP synthase (cGAS), a cytosolic DNA sensor that activates the innate immune system. cGAS and its downstream target STING are believed to form an essential node between cancer cells and the immune microenvironment, as CIN often coincides with cytosolic DNA [14,15].

In this review, we will discuss recent findings that link CIN to the cGAS-STING axis and what is known of the underlying molecular mechanism. We will highlight how pathway activation can affect cancer cells’ viability, and under which conditions this is beneficial or detrimental for cancer cells in relation to CIN. In addition, we will assess how these consequences could be exploited in therapy and how this would influence the treatment window of cancer patients.

## 2. Upstream Triggers of the cGAS-STING Pathway

Discovered as a signaling cascade that detects viral DNA in the cytoplasm [16], more recently cGAS-STING signaling has received a lot of interest from the CIN field. As the cGAS pathway is activated through binding of cGAS to cytoplasmic double-stranded DNA (dsDNA) [16], it does not only detect viral DNA, but also cytoplasmic dsDNA in micronuclei, an extra-nuclear body that contains whole chromosomes or chromosome fragments that failed to be included in the main nucleus after mitosis (Figure 1). Only when the membrane of the micronucleus ruptures, dsDNA becomes accessible for cGAS binding [14,15] (Figure 1). Upon binding of dsDNA, cGAS undergoes a conformational change that enables cGAS to produce cGAMP via its C-terminal domain by cyclizing ATP and GTP [16,17,18] (Figure 1). While it has been reported that cGAS also binds to single stranded DNA (ssDNA) and double stranded RNA (dsRNA) [19], this binding does not cause a conformational change in cGAS, and therefore does not yield cGAMP production [20]. Additionally, cGAS seems to be activated by dsDNA in a length dependent manner, as human recombinant cGAS is more efficiently activated by larger DNA fragments (kilobase range) when compared to shorter fragments. The efficiency of cGAS activation also dictates the downstream IFN production, with longer DNA fragments to be more immunostimulatory [21].

Downstream of cGAS, cGAMP binds to STING (STimulator of INterferon Genes, Figure 1). Under unperturbed conditions, STING localizes to and is retained in the ER through its interaction with stromal interaction molecule 1 (STIM1, indicated as * in Figure 1) [22]. Upon the binding of cGAMP, STING undergoes a conformational change and translocates to the Golgi apparatus [23] (Figure 1). When localized to the Golgi, STING is palmitoylated at cysteine residues 88 and 91, which catalyzes the recruitment of TBK1 and its downstream component IRF3 [24]. STING-TBK1 binding leads to the phosphorylation of IRF3^S386^ and its dimerization, followed by IRF3 translocation into the nucleus to drive the transcription of interferon-β, mediating the type I IFN response and other IRF3 target genes [25] (Figure 1). In addition, IRF3 nuclear translocation leads to STING degradation in endolysosomes to switch the signaling cascade back off [26]. Parallel to IRF 3 and type I IFNs activation, cGAS-STING pathway activation also leads to the activation of canonical and non-canonical NF-kB (nuclear factor kB) in cancer cells, as further discussed below [27] (Figure 1). While both these pathways have strongly been implicated with regulation of cell death and immune activation, as discussed below, further work is required to better understand the interdependencies between these downstream branches.

## 3. Downstream Effects of the cGAS-STING Pathway: The Innate Immune System Response

### 3.1. Type I Interferon (IFN) Response and Signal Transducer and Activator of Transcription (STAT) Response

The key output of the cGAS-STING signaling cascade is the upregulation of type I IFNs (Figure 1). Type I IFNs are a family of monomeric cytokines consisting of 14 well-defined IFNα subtypes, IFNβ, and the more poorly understood cytokines IFNε, IFNκ, and IFNω [28]. Type I IFNs are known for two major roles: (1) the regulation of the innate immune response and (2) the activation of the adaptive immune system. In a cancer context, type I IFNs are mostly known for their cell-intrinsic tumor suppressing role in premalignant cells through cell-intrinsic upregulation of p53, negative regulation of cell proliferation, induction of apoptosis, and cell extrinsic activation of immune surveillance for tumor cell clearance [29].

The canonical type I IFNs include IFN-α and IFN-β. The binding of IFN-α or IFN-β to their respective receptors (IFNAR or IFNBR) leads to the activation of Janus kinase 1 (JAK1) and tyrosine kinase 2 (TYK2), and downstream recruitment, dimerization, and nuclear translocation of STAT (Signal Transducer and Activator of Transcription) proteins, as schematically shown in Figure 2 [29]. STAT family member STAT1 binds to INF gamma-activated sequences (GASs) as a homodimer to promote the transcription of pro-inflammatory genes, such as IRF1 and CXCL9 [30] (Figure 2). Conversely, STAT3 homodimers are formed following the activation of JAK-JAK receptors, which in turn act as suppressors of pro-inflammatory genes through an unknown transcriptional suppressor [31] (Figure 2).

The cGAS-STING-driven type I IFN response activates the innate immune system and, in particular, natural killer (NK) cells, which enhances their cytotoxic activity [32]. Impaired type I IFN signaling thus impairs NK cells’ cytotoxic activity as the IFN signaling downstream cytokines, such as IL-15 and IL-12, are required to prime NK cells towards their full competence [33]. Parallel to STAT1 activation, cGAS-STING activity also promotes STAT3 activity, which counteracts STAT1-mediated activation of NK cells in a negative feedback loop. As such, STAT3 promotes an immunosuppressive tumor microenvironment, which can be partly attributed to reduced NK cell activity [34]. Furthermore, STAT3 activation reduces tumor cell sensitivity for NK-mediated killing via alteration of NK-activating ligands such as NKG2D [35]. Additionally, STAT3 activity decreases the migration of various immune cells to the tumor microenvironment, including NK cells, T cells, neutrophils, and macrophages, further contributing to an immunosuppressive tumor microenvironment [36]. Conversely, the inhibition of STAT3 activity using STAT3 inhibitors elevates the level of chemoattractant chemokines, including CCL2, CCL9, CCL12, and CCL17, and enhances tumor cell sensitivity to NK-mediated lysis [37]. Together, these features make STAT3 a promising therapeutic target.

### 3.2. Nuclear Factor kB (NF-KB) Response

The NF-kB family consists of five subunits: RelA, RelB, c-Rel, p50, and NF-kB p52, modulates the immune response, cell survival, and proliferation [38], and can be activated by cGas signaling (Figure 1). While the canonical NF-kB pathway primarily depends on a RelA/p50 dimer, non-canonical NF-kB signaling relies on RelB and p52. Non-canonical NF-kB pathway activation is slow, being induced by tumor necrosis factor receptor (TNFR) family members ligands, and persistent when activated. The processing of p100 is a crucial step for non-canonical NF-kB pathway activation and is a tightly regulated process. It results in the production of non-canonical NF-kB p52 and nuclear translocation of NF-kB p52 and RelB, leading to transcriptional activation of a large number of genes and cytokines, including CXCL1, CD44, and HIF-2α [39,40,41]

Aberrant non-canonical NF-kB activity has been associated with multiple malignancies, including multiple myeloma, pancreatic cancer, prostate cancer, glioblastoma, and breast cancer [42]. For instance, RelB expression in prostate cancer cells promotes epithelial-to-mesenchymal transition (EMT) and negatively correlates with the survival of prostate cancer patients [43]. RelB expression is also increased in ER negative breast cancer when compared to the less aggressive ER positive breast cancer [44]. In addition, RelB and its p52 binding partner are also known to regulate the activity of a group of nucleic acid editing enzymes (APOBECs, [45]). The increased activity of APOBECs is believed to be tumor promoting, as it induces mutations at cytosine residues with a thymine, for instance through mutation, which increases RelB/p52 activity in breast and ovarian cancers [46].

## 4. The Bright Side: cGAS as a Suppressor of Chromosomal Instable Tumors

Chromosomal instability can, for instance, through the formation of micronuclei, lead to cytoplasmic DNA and hence the activation of the cGAS-STING signaling axis [14,15,27]. The resulting micronuclei can contain whole chromosomes as well as chromosome fragments, depending on the type of CIN [47]. When micronuclei rupture, DNA becomes accessible for cGAS binding and thus cGAS-STING pathway activation, ultimately driving the type I IFN response and immune surveillance, as described above [14,15] (Figure 3A).

Cancer immune surveillance can be triggered through cell intrinsic and cell extrinsic cues. Cell intrinsic cGAS-STING activation typically results from chromosome segregation errors or DNA damage that lead to cytoplasmic DNA accumulation, including micronuclei. When micronuclei rupture, cGAS is activated followed by the downstream activation of type I IFN (Figure 3A). This will trigger the innate immune system as described above. Indeed, work from Santaguida et al. elegantly demonstrates that complex karyotypes induced in normally diploid retinal pigment epithelial (RPE1) upregulate the cGAS-STING pathway. The upregulation of the cGAS-STING pathway increases the expression of a number of cytokines, including IL-6 and CCL2 (Figure 3A). Indeed, RPE-1 cells with complex karyotypes are cleared by cultured natural killer cells (NK-92 cells) more efficiently in comparison to its wild type counterpart as a result of cGAS-STING pathway activation and the downstream type I IFN response [48].

cGAS-STING activity can also be triggered in a cell non-autonomous manner, for instance through uptake of cancer cells by antigen presenting cells (dendritic cells and macrophages [49], Figure 3B, upper panel). In this setting, DNA from the engulfed cells triggers the activation of the cGAS-STING signaling cascade. As shown for cell-intrinsic cGAS activation, cGAS cell-extrinsic activation also promotes the priming, expansion, and recruitment of tumor-specific T-cells through type I IFNs and their downstream pro-inflammatory molecules [50,51] (Figure 3A, B). In addition to that, cGAMP that is produced by cancer cells is secreted and taken up by dendritic cells, either through pinocytosis or endocytosis. This further boost STING and its downstream pathway activation in the dendritic cells through the production of IFNβ in a feed forward manner.

In addition to modulating the innate immune system, cGAS also impinges on the adaptive immune system (Figure 3A). This followed, for instance, from observations that STING-deficient mice display decreased the infiltration of tumor-specific CD8^+^ cells and decreased levels of the T cell activation factors IL-12, CD86, and CD40 [52]. In this setting, cGAS-STING activates dendritic cells that prime CD8^+^ T cells through cross-presentation. Furthermore, ectopic administration of cGAMP in vivo has been shown to elevate the infiltration of tumor-specific CD4^+^ T-cells, macrophages, dendritic cells, and CD8^+^ T-cells, leading to the increased production of type I IFN and ISGs. Finally, type I IFNs are also responsible for the upregulation of Th1 chemokines, such as CXCL10 (Figure 3A), an important chemokine for the homing of antigen-presenting cells and trafficking of CD8^+^ T-cells [53]. Together, these observations strongly suggest that cGAS-STING signaling also has a role in modulating the adaptive immune system.

Moreover, cGAS also acts as tumor suppressor independent to immune system signaling. For instance, a non-canonical role of cGAS was recently reported as an instigator of mitotic cell death during prolonged mitotic arrest [54] (Figure 3A). In this setting, cGAS activity triggers gradually accumulating levels of phosphorylated IRF3 that promote mitotic cell death independent of type I interferon induction, as evidenced by the lack of transcription of downstream type I IFNs proteins, such as CXCL10, IFNB1, and IRF1. Instead, cGAS activity promoted the suppression of Bcl-xL dependent inhibition of Mitochondrial Outer Membrane Permeabilization (MOMP), caspase activation and protein substrate cleavage, which lead to mitotic cell death upon prolonged arrest.

Given that the cGAS-STING pathway plays an active role in eliminating tumor cells, one would assume that the alleviation of the pathway would provide a powerful means for cancer cells to duck immune surveillance, particularly when displaying a CIN phenotype. However, large scale cancer sequencing datasets, such as The Cancer Genome Atlas (TCGA), reveal that genes encoding cGAS or STING are rarely mutated when assessing more than 10,000 tumors. Only 0.6% of the tumors have mutations in cGAS and 0.5% in STING with only half of these mutations leading to amino acid changes [55]. Copy number loss of cGAS or STING is also rare, instead, amplifications are much more common [56].

To further complicate matters, cGAS-STING levels in tumor cells vary greatly from one tumor type to another. In colorectal cancer, the low expression of cGAS and STING is often associated with more advanced stages of tumors. Similarly, the STING mRNA levels are reduced in progressed gastric cancers. Yet, in breast, prostate, and head-and-neck tumors, cGAS and STING expression is increased when compared to the healthy tissues [27,57,58,59,60]. The increased expression might result from epigenetic deregulation, as breast, pancreatic, head-and-neck, and lung cancer frequently show decreased methylation at cGAS and STING promotor regions. Therefore, apparently, despite the predicted advantage of hiding from immune surveillance, the inactivation of the cGAS-STING signaling axis is not a universal oncogenic path [55].

A part of the answer to this apparent paradox might be that the activation of the cGAS-STING pathway in cancers does not always lead to downstream immune signaling. For example, the breast cancer cell line MDA-MB-231 has a functional cGAS-STING axis, but pathway activation does not yield a type I IFN response or any other immune signaling, which suggests that cancer cells have inactivated the signaling route more downstream [27]. Several mechanisms might explain this:

A first possible mechanism that could explain active cGAS-STING without an immune response is the inactivation of proteins more downstream in the pathway closer to the actual immune signaling. For instance, the downstream effectors such as STAT1, JAK, and TYK2 are important triggers of the type I IFN response. Indeed, the oncogenic tyrosine kinase NPM-ALK has been reported to bind to and phosphorylate STAT1 [61], triggering its proteasome-dependent degradation. Additionally, promoter methylation and thus reduced expression of STAT1 is common in head and neck cancer [62]. Furthermore, the activation of STAT1 negative regulators (i.e., SOCS (suppressor of cytokine signaling), PTPs (protein tyrosine phosphatases), and PIAS (protein inhibitor of activated STAT)) is another route that cancer cells take to block the type I IFN response. Indeed, STAT1 inhibitors SOCS1 and SHP1/2 (component of PTPs) are also commonly hyperactivated in breast cancer [63], and elevated expression of PIAS1 has been observed in prostate and breast cancers [63,64,65]. Conversely, STAT3, the yang to STAT1′s yin, is a *bona fide* oncogene, which provides another route to counteract STAT1 activity. Indeed, STAT1 and STAT3 ratios have long been recognized as a critical balance to regulate cell growth [66].

A second possible mechanism to silence cGAS-STING signaling downstream might relate to p38-MAPK stress signaling. When the p38 MAPK-kinase stress response pathway is activated alongside with cGAS, for instance, as a result of CIN, the cells fail to activate type I INFs [67]. A possible mechanism for this negative interaction is that p38 MAPK signaling catalyzes phosphorylation of the deubiquitinating enzyme USP21, which in turn inhibits STING activity [68]. This would thus prevent type I INF activation without deregulating cGAS or STING expression levels, and thus be another mechanism for cancer cells with a CIN phenotype to circumvent immune surveillance.

Whichever the underlying mechanism might be, one key conclusion is that tumors tend to block IFN signaling preferentially downstream of cGAS and STING expression, which suggests that cGAS and STING might have IFN-independent functions that are beneficial for cancer cells.

## 5. The Dark Side: cGAS-STING Signaling is Required for CIN Tumors

As discussed above, the frequent activation of cGAS and/or STING signaling in cancer strongly suggests an oncogenic role for this pathway. One possible explanation comes from a recent study that implicates cGAS and STING with a role in metastasis in CIN tumors ([27], (Figure 3A). This study started from the observation that metastatic breast cancer clones often display more chromosomal instability when compared to the primary tumor. For this, MDA-MB-231 breast cancer cells with or without a CIN phenotype were transplanted into immunocompromised mice and metastatic events were tracked while using a bioluminescence reporter to determine whether CIN would accelerate metastasis. Mice that received MDA-MB-231 cells exhibiting a CIN phenotype displayed an average survival of 70 days before succumbing to metastasis-related complications, while the mice transplanted with MDA-MB-231 cells without a severe CIN phenotype survived much longer (up to 207 days) and had lower metastatic burden when sacrificed. In addition to giving an increased metastatic burden, MDA-MB-231 CIN^high^ cells had increased expression of EMT genes as compared to MDA-MB-231 CIN^low^ cells. Further analysis revealed a strong correlation between inflammation-related genes, CIN signature genes, and EMT genes, which suggests that the link between CIN and EMT might be mediated through an inflammatory mechanism. Indeed, when metastasis latency and burden were compared between MDA-MB-231 CIN^high^ cells with and without STING knockdown as a modulator of the immune system, they found that functional STING was required for efficient metastasis. The EMT phenotype and resulting metastasis critically relied on the activation of the non-canonical NF-kB pathway (RelB/p52) as triggered by cGAS-STING signaling. Conversely, STING-depleted cells had reduced nuclear RelB and downregulated expression of EMT genes, indicating that inactivating components in non-canonical NF-kB signaling can prevent metastasis, an important clinical implication that we will further discuss below.

In addition to promoting metastasis, cGAS has also been reported to promote tumorigenesis by modulating the DNA damage response through the suppression of homologous recombination-mediated DNA repair [69] (Figure 3B, lower panel). In their study, Liu et al. find that DNA damage triggers Blk-mediated phosphorylation of cGas^Y215^, leading to cGAS nuclear translocation. Nuclear cGAS is recruited to the site of the double-stranded break, where it exerts a suppressive effect exclusively on homologous recombination (HR)-mediated repair, while leaving the error prone non-homologous end joining (NHEJ) pathway unaffected. The suppressive effect of cGAS on HR repair is mediated through an interaction with PARP1 through PAR that prevents the formation of the PARP1-Timeless complex. The observation that cGAS modulates HR is further supported by another study [70] that demonstrates that cGAS inhibits homology-directed repair in a STING-independent manner. Interestingly, the inhibition of ATM, a key player in the HR pathway, rescues the increase of DNA breaks caused by cGAS, while the inhibition of DNA-PK, an effector in NHEJ, does not, providing further proof for a role of cGAS in HR, but not in NHEJ. The proposed mechanism is that cGAS interferes with Rad51-mediated DNA strand invasion by binding to the double strand template used by Rad51 filaments for repair, thus effectively inhibiting HR. These observations suggest that cGAS can act as a suppressor of homology-directed repair, thus decreasing the efficiency of DNA repair thus promoting tumorigenesis.

Together, these data reveal a darker side of cGAS-STING signaling, as a promotor of metastasis and an inhibitor of homology-directed DNA repair, providing a possible explanation for the frequent activation of cGAS-STING signaling in cancer.

## 6. Therapeutic Implications of Altered cGAS-STING Signaling

The dual role of cGAS in chromosomal instability as a tumor suppressor through its activation of the innate immune system and, conversely, as a tumor promoter by inhibiting DNA repair and promoting metastasis might come with important implications for cancer therapy.

Even though the cGAS-STING signaling axis has only recently been identified and many aspects of this pathway need further investigation, many of the ‘classic’ chemotherapeutic and radiotherapeutic approaches to eradicate cancer heavily rely on this signaling route (Figure 4A). For instance, etoposide has previously been shown to induce the expression of inflammatory genes, such as IFNβ, IFNA4, and IFI16 [71]. Similarly, dimethyloxoxanthenyl acetic acid (DMXAA), another chemotherapeutic agent, induces IFNβ and primes CD8^+^ cells in a STING-dependent manner [72]. Cisplatin treatment has also been reported to activate the cGAS-STING pathway, which boosts the expression of type I interferon genes, such as CXCL9 and CXCL10 [73]. Both of the chemokines have been reported to recruit antigen presenting cells and T cells to the tumors.

Similar as described for chemotherapeutics, radiotherapy will trigger cGAS-STING signaling through the generation of neo-epitopes that will activate dendritic cells, and through the accumulation of cytoplasmic DNA that will directly trigger cGAS-STNG in the cancer cells (Figure 4A). The intimate relationship between cGAS, DNA damage and the immune system opens up possibilities to exploit radiotherapy-induced DNA damage to trigger the immune system to clear even more cancer cells [15]. Indeed, combining gamma radiation with immune checkpoint inhibitors appears to be a very powerful cGAS-STING dependent approach for treating metastatic cancer in a mouse model for melanoma [15]. In addition to the more traditional anti-cancer therapies, more recently-developed targeted therapies, such as DNA damage response (DDR) inhibitors (e.g., PARP inhibitors, Chk inhibitors, DNA-PK inhibitors, ATR inhibitors), also activate cGAS-STING signaling and prompt the downstream immune components entailed [74] (Figure 4A). Additionally, these compounds provoke cytosolic DNA fragments that are recognized by cGAS following the activation of the pathway, emphasizing the strong synergy between DNA damaging agents and cGAS-STING signaling in clearing cancer cells (Figure 4A).

As eluded to above, cGAS-STING dependent anti-tumor immunity is of crucial importance for the success of immunotherapy, such as anti PD-1 and anti-PD-L1 immunotherapy (Figure 4A). This is exemplified by a study showing that the PD-L1 inhibitors failed to give any anti-tumor effect in in cGAS deficient mice [51]. Importantly, introducing cGAMP downstream of cGas rescued this effect, providing strong evidence that cGAS-STING signaling is required for the therapeutic effect of immune checkpoint inhibitors. Similarly, the STING agonist 2′3′-c-di-AM (PS) 2 (Rp, Rp) acts synergistically with the immune checkpoint inhibitors anti-PD-1 and anti-PD-L1 in clearing tumors [75]. Additionally, in this setting, the STING agonist promoted tumor infiltration of cytotoxic T cells, which was fully dependent on anti PD-1 or anti-PD-L1 treatment. These findings provide substantial proof for a synergy between a chemotherapeutically-induced cGAS response and immune checkpoint blockade in cancer therapy (Figure 4A).

Moreover, the recently-discovered non-canonical role of cGAS in promoting mitotic cell death during a prolonged mitotic arrest [54] described above opens up yet another therapeutic implication. Several commonly-used chemotherapeutic drugs (Vincristine, taxanes, etc.) act by prolonging mitosis through interfering with the mitotic spindle dynamics (Figure 4A). Indeed, Paclitaxel shows increased therapeutic efficiency in breast cancer cells that express high levels of cGAS. Therefore, high cGAS expression might be a good predictor of clinical outcome when treating with microtubule poisoning drugs, as cGAS expression would promote cell death in cells with prolonged mitotic arrest.

Our rapidly increasing understanding of the cGAS-STING signaling axis also leads to a better understanding of which genetic contexts render cells more or less reliant on this pathway (Figure 4A, B). For example, work from Heijink et al. shows that BRCA2 deficient breast cancer cells are more sensitive to TNFα due to cGAS-STING activation [76]. BRCA2 deficient tumors have impaired homologous repair, which leads to increased numbers of micronuclei, cGAS-STING activation, and a downstream interferon response. The interferon response triggers activation of JNK and ASK1 kinases, tipping the balance of TNFα-induced NF-kB pro-survival signaling towards pro-apoptotic signaling, thus rendering these cells sensitive to TNAα. Similarly, ribonuclease H2 has been reported to induce cGAS-STING signaling. Mice harboring a Rnaseh2b^A174^ mutation that show upregulation of interferon stimulated genes (ISG) in various tissues as a result of cGAS-STING activation [14] are therefore also expected to be very sensitive to TNFα, similar to BRCA2 deficient tumor cells. These observations suggest that, while tumors that have cGAS-STING activation might have found ways to circumvent the immune system, they can still be targeted by modulating the cGAS-STING pathway upstream to trigger cell death.

However, activation of the cGAS-STING pathway does not always confer anti-tumor effects. As discussed above, cGAS-STING signaling also facilitates metastasis in chromosomally instable cells. Therefore, for cells exhibiting a CIN phenotype, the inhibition of cGAS or STING might a better therapeutic option. Recently developed small molecule inhibitors for cGAS and STING can be used to test this (Figure 4B). However, such compounds will also block innate immune surveillance [77,78] and, therefore, while such small molecule inhibitors have the potential to block CIN tumors from becoming metastatic [27], cGAS/STING inhibition might come at the price of local immune suppression and thus impaired tumor clearance. Therefore, further work is required to determine how this plays out in vivo. Alternatively, as non-canonical NF-kB has been described to drive EMT and metastasis downstream of cGAS-STING signaling, targeting essential components of this pathway is an option that be further explored. However, more studies on the side effects of targeting this pathway are pivotal, given the importance of NF-kB signaling.

From the work described above, an overall picture is emerging that the modulation of cGAS-STING signaling can potentially be used to target cancer cells, but that the type of modulation (activating or inactivating) is highly context dependent. While in genome-stable tumors ectopic induction of the cGAS-STING pathway might be beneficial, as it would trigger type I IFNs and canonical NF-Kb pathway promoting apoptosis, T-cells infiltration, and NK cells killing [56] (Figure 4A,B), cGAS-STING signaling is constitutively activated in CIN tumors, and therefore the latter type of cancers have presumably found ways to circumvent downstream immune surveillance. Therefore, the inhibition of the cGAS-STING axis might be a better approach to target CIN tumors, as this would block the activation of non-canonical NF-kB signaling, thus preventing an EMT phenotype and thus impairing metastasis. While these clinical implications are still mostly hypothetical, together these findings strongly suggest that the CIN status of a tumor could become a key determinant in treatment stratification.

## 7. Conclusions

Chromosomal instability has beneficial consequences for tumors, as it fuels tumors with the potential to adapt their cells’ karyotypes, thus allowing for individual cancer cells to acquire new features to adopt a more malignant fate [2,3]. Conversely, constant karyotype changes will also lead to reduced cellular fitness for individual cells, which explains why cancer karyotypes recur between tumors and are still reasonably constant when assessed at the population level, despite ongoing CIN phenotypes [3]. The cGAS-STING signaling axis adds another layer of complexity to the role of CIN in cancer. While the activation of cGAS-STING signaling in tumors might provide a powerful means to mobilize immune surveillance and thus lead to tumor clearance, CIN tumors will likely have adapted to sustained cGAS-STING signaling and thus have become insensitive. Intriguingly, CIN tumors even seem to employ sustained cGAS-STING signaling to increase DNA damage and help cells adopt a metastatic phenotype. Therefore, when combined with cytotoxic drugs, the inhibition of cGAS-STING signaling could provide a powerful means for improving the outcome of CIN cancers. However, more work is needed to further resolve the molecular players of this intriguing pathway and to determine in which clinical settings activation or inhibition of this pathway can be employed in the fight against cancer.

## Figures and Tables

**Figure 1 cells-08-01228-f001:**
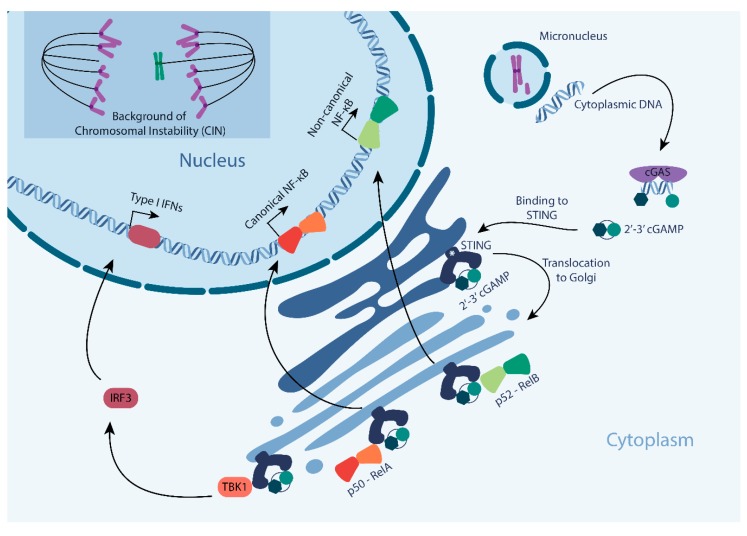
Cyclic GMP AMP synthase-STING (cGAS-STING) pathway signaling as activated by genomic instability. Cells with ongoing chromosomal instability (CIN) produce structures that contain genomic DNA, known as micronuclei. Upon the rupture of its membrane, dsDNA contained in a micronucleus is exposed to the cytoplasm and bound by cGAS. The binding of cGAS to dsDNA triggers the production of its secondary messenger, cGAMP that in turn binds and activates STING. Activated STING kinase translocates from the ER to the Golgi and phosphorylates its downstream targets, initiating (1) a TBK1-IRF3-type I interferon response, (2) a canonical NF-kB response, or (3) a non-canonical NF-kB response. * indicates stromal interaction molecule 1 (STIM1).

**Figure 2 cells-08-01228-f002:**
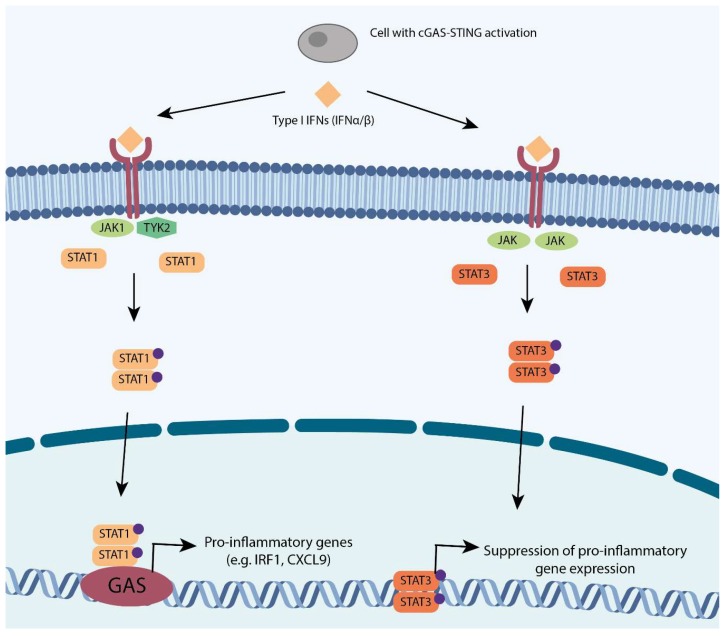
The two branches of type I interferon: STAT1 and STAT3. The STAT1 branch: IFN α/β activate JAK1-TYK2 receptors which in turn phosphorylate STAT1, causing its dimerisation. STAT1 dimers travel to the nucleus to drive pro-inflammatory genes. The STAT3 branch: The same substrates also activate JAK-JAK receptors that in turn phosphorylate STAT3 catalyzing its dimerisation. STAT3 dimers translocate to the nucleus to suppress of pro-inflammatory genes. Phospho-groups are indicated in purple.

**Figure 3 cells-08-01228-f003:**
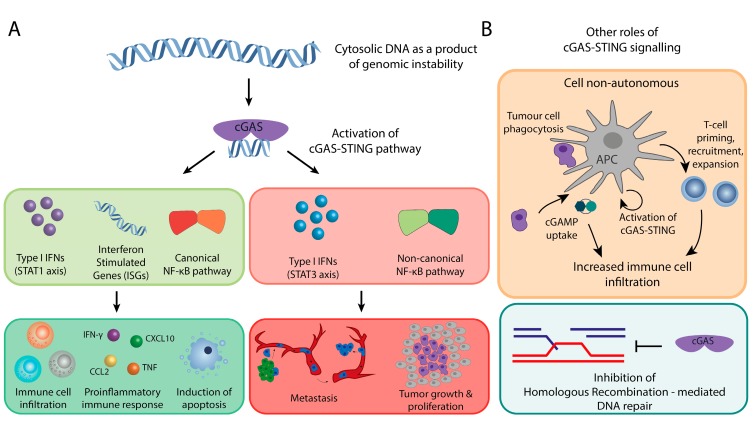
The two faces of cGAS-STING signaling in cancer. (**A**) cGAS-STING acts as a tumor suppressor by initiating type I interferon signaling (STAT1 axis) and canonical NF-kB signaling. These signaling axes promote immune cell infiltration and cell death in tumors through modulation of transcription of pro-inflammatory and apoptosis signaling molecules. Conversely, it also triggers the other side of type I interferon signaling, the STAT3 axis, and the non-canonical NF-kB pathway to promote metastasis and tumor growth. (**B**) Other roles of cGAS-STING signaling: (1) a cell non-autonomous role in which cancer cells’ DNA is engulfed by antigen presenting cells (APC) leading to activation of cGAS-STING signaling and cGAMP production in APCs. cGAMP is secreted by the APCs further stimulating immune infiltration (upper panel). (2) Nuclear cGas can also act as an inhibitor of homologous recombination-mediated DNA repair (lower panel).

**Figure 4 cells-08-01228-f004:**
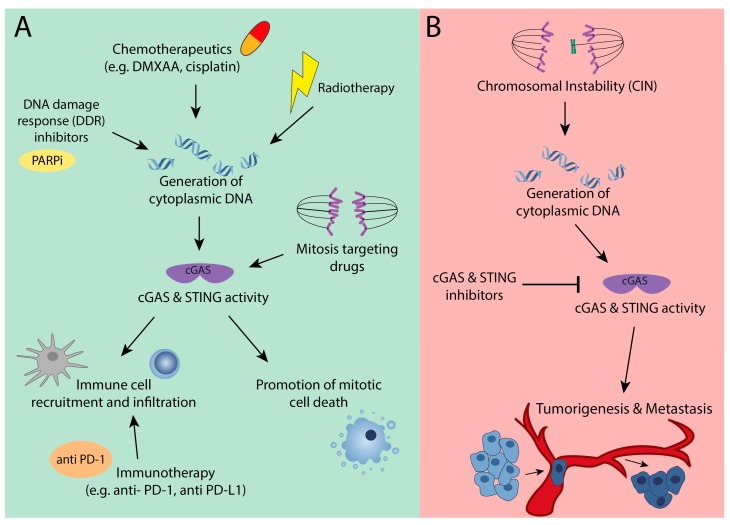
Context dependent cancer therapeutic strategies when targeting cGAS or STING. (**A**) Targeting the tumor suppressor role of cGAS-STING through activation of cGAS-STING signaling by different means to generate cytoplasmic DNA (radiotherapy, chemotherapy, DNA damage response (DDR) inhibitors, and others) promotes immune cell infiltration into tumors and cell death in mitosis. (**B**) Targeting the oncogenic role of cGAS-STING signaling through cGAS or STING chemical inhibitors could delay tumor progression and suppress metastasis.

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
