# Peer review of "The cGAS Paradox: Contrasting Roles for cGAS-STING Pathway in Chromosomal Instability"

_cells, 2019, doi:10.3390/cells8101228_

Round 1

Reviewer 1 Report

In this review, Hong et al. discuss the recent findings on the role of cGAS-STING in chromosomal instability. This is an excellent review, extremely well written and referencing, very precisely, all the recent work in the field. Thus, the review is ready to be published in its current form.

I have only one suggestion for the authors. In the introduction, line 27: I do not agree with the statement "CIN is the process that leads to aneuploidy", since cells can be aneuploid without being chromosomally unstable. An example of this is Ts21 in Down Syndrome which is not the result of CIN. Thus, I would rather say that CIN is a condition in which cells mis-segregate their chromosomes with a high frequency and this leads to aneuploidy. 

Author Response

We thank this reviewer for the assessment of our review and kind words. We agree with the reviewer's suggestion and have changed the text accordingly.

The text now reads:

CIN is a condition as a result of which chromosome missegregation happens at increased frequency while aneuploidy is a statedescribing the actual abnormalities in chromosome copy numbers. While CIN and aneuploidy often co-occur in tumors (Storchova, 2018)aneuploidy can also exist in the absence of CIN, a condition known as stable aneuploidy (Storchova & Kuffer, 2008). An example of stable aneuploidy occurs for instance in Down syndrome patients whose cells all harbor one extra copy of chromosome 21. Similarly, some cancers display high-grade aneuploidy without a CIN phenotype and are thus defined to be stably aneuploid (Storchova & Kuffer, 2008). 

Reviewer 2 Report

Recent work has uncovered the intriguing link between chromosome instability (CIN) and cGAS signaling, which may be a critical signaling mechanism between the tumor and its microenvironment.  In this review, the authors discuss this most recent work and put it in the larger context of what is known about cGAS signaling and tumorigenesis.  This review is therefore timely and important. 

Overall, this is up to date and provides good perspectives on the field.  I love that it starts out with a very clear description of the difference between CIN and aneuploidy, as so many scientists do not understand the clear distinctions, which leads to confusion in the literature. When it gets into more detailed signaling, it is not as clear as it could be.  I think this is largely because the figures are not used effectively.  The figures are quite simplistic relative to the complex discussions of the molecular pathways in the text, and the text often refers to the figure in places where the figure does not actually illustrate the point being discussed.  Therefore, while I support eventual publication of this timely piece, there would be a significant increase in clarity to the reader if the authors rethought the figures and focused on revising figures to coordinate better with the text as written.  I have highlighted a few specific examples of this below. 

In Figure 1, the text discusses additional proteins that are not in the figure, which is confusing to the reader. Please amend the figure to include the additional proteins- ie STIM1 and if something is not going to be shown, then please indicate in the legend (or text).  Also, as the section develops, there are a large number of integrated signaling pathways that are discussed.  An additional figure indicating the major and minor pathways would provide a clearer picture of the signaling pathways involved.  Otherwise it is just molecular soup to someone outside the field.  The authors refer to Figure 2 when discussing the cell non-autonomous stimulation of cGAS-STING, but this is not illustrated in the figure. In addition, the right arm of the figure is barely described in the legend. On line 294, the authors are making the concluding comments about the role of cGAS as a suppressor of homology-directed repair and refer to Figure2, which does not illustrate anything about DNA repair mechanisms. The ideas discussed with regards to how the cGAS pathway could be targeted in tumors are interesting ideas. A figure to highlight the main take home points of these hypotheses would be more thought-provoking for the reader. 

Author Response

We thank this reviewer for the assessment of our manuscript and are pleased that the reviewer is in favor of publication pending some changes. We agree with the reviewer that we could make better use of figures. We have therefore added two figures (Figures 2 and 4) and extended Figure 3 (previous figure 2). We also refer better to the figures in the text. We think that these additional figures significantly improve the manuscript.

Reviewer 3 Report

Hong et al., in this review article, tried to address the role of cGAS/STING pathway in chromosomal instability. They have explained detailed mechanism of cGAS/STING pathway and provided evidences of  the contradictory role of cGAS/STING pathway in context of cancer. However, there are multiple reviews that has already addressed this issue (for example: Bose D. Int J Mol Sci. 2017). Though they have provided some literature evidences on the role of cGAS/STING pathway in promoting CIN tumors, there is no direct evidence of its inhibitory role in context of CIN tumors. Rather, they have described the anti-cancer potential of cGAS/STING pathway.

This review article is misleading as it does not justify its title and also does provide any new information. Therefore, I do not think this has the potential to be published in the current form. 

Author Response

We thank this reviewer for the assessment of our manuscript. We disagree with the reviewer that the title of our manuscript is misleading: we discuss the pro's and con's of cGAS-STING signaling in the context of CIN cancers in our manuscript and this is what we state in the title. We agree that there are more reviews that discuss this topic, but we still want to give our interpretation of the studies that have been published lately. We believe that our extensive review and schematic overview of the pathways and recently published studies have added value for people who are new to this field. 

We hope that this reviewer still appreciates the amendments to the text and figures as suggested by the other 3 reviewers, who seem to be more in favour of publication of this review pending minor amendments. 

Reviewer 4 Report

The review „The cGAS paradox: contrasting roles for cGAS-STING pathway in chromosomal instability” by Hong C. et al. provides comprehensive summary of our current understanding of the role of cGAS-STING pathway in tumour development, pointing out the important role of this signalling axis for immune system response, promotion of metastasis, and DNA damage repair by homologous recombination. The potential implication for future studies and possible therapeutic interventions of cGAS-STING pathway are discussed in sufficient detail.

Overall, the manuscript is well written and might attract a broad range audience.

In my opinion, it might be accepted after some minor corrections.

Minor points:

Lane 27 – The text is formatted „bold“. Is there any reason for that?

Lane 45 – Use a full name for TCGA abbreviation here (The Cancer Genome Atlas).

Lane 49 – Should be corrected to: …cGAS and its downstream protein, STING…

Lane 71 – Delete the repeating words (DNA larger DNA fragments (kilobase range) compared to shorter DNA).

Lane 381 – Make correction of the phrase as follows: “…a key determinant in treatment stratification.”

Author Response

We appreciate the detailed assessment of our manuscript by this reviewer and we have amended the text according to this reviewer's suggestions.

We left the bold text in lines 27 and 28 and we want to emphasize the words condition and state to highlight the differences between CIN and aneuploidy. However, if the reviewer disagrees with this/finds this confusing, we are happy to remove the bold formatting, as other readers might feel the same.

Round 2

Reviewer 3 Report

 As mentioned before, there is no direct evidence of its inhibitory role in context of CIN tumors. The authors have just described the general anti-cancer role of cGAS/STING pathway. Therefore, I still think that the title is misleading and also this review article does provide new information. The authors have only focused on CIN tumors, that limits the scope/reader of the review. And then, there is no direct evidence of cGAS/STING pathways inhibitory role in CIN. 

Thus, I still think that the article should not be considered for publication.